# A versatile informative diffusion model for single-cell ATAC-seq data generation and analysis

**Lei Huang** *
City University of Hong Kong
Massachusetts Institute of Technology

**Lei Xiong** *
Massachusetts Institute of Technology

**Na Sun**
Massachusetts Institute of Technology

**Zunpeng Liu**
Massachusetts Institute of Technology

**Ka-Chun Wong**
City University of Hong Kong

**Manolis Kellis** †
Massachusetts Institute of Technology

## Abstract

The rapid advancement of single-cell ATAC sequencing (scATAC-seq) technologies holds great promise for investigating the heterogeneity of epigenetic landscapes at the cellular level. The amplification process in scATAC-seq experiments often introduces noise due to dropout events, which results in extreme sparsity that hinders accurate analysis. Consequently, there is a significant demand for the generation of high-quality scATAC-seq data *in silico*. Furthermore, current methodologies are typically task-specific, lacking a versatile framework capable of handling multiple tasks within a single model. In this work, we propose ATAC-Diff, a versatile framework, which is based on a latent diffusion model conditioned on the latent auxiliary variables to adapt for various tasks. ATAC-Diff is the first diffusion model for the scATAC-seq data generation and analysis, composed of auxiliary modules encoding the latent high-level variables to enable the model to learn the semantic information to sample high-quality data. Gaussian Mixture Model (GMM) as the latent prior and auxiliary decoder, the yield variables reserve the refined genomic information beneficial for downstream analyses. Another innovation is the incorporation of mutual information between observed and hidden variables as a regularization term to prevent the model from decoupling from latent variables. Through extensive experiments, we demonstrate that ATAC-Diff achieves high performance in both generation and analysis tasks, outperforming state-of-the-art models.

## 1 Introduction

Assay for Transposase Accessible Chromatin with sequencing (ATAC-seq) data has shed light on genome research to dissect gene regulatory landscapes and cellular heterogeneity. Particularly, single-cell ATAC-seq (scATAC-seq) can be harnessed to probe the chromatin accessibility profile at single-cell level to reserve the diversity of cell types in a heterogeneous tissue, which can reveal important genomic elements for transcription factor binding and regulating downstream gene expression, particularly at distal non-coding regulatory regions[1, 2]. However, the analysis of such data poses challenges due to the high levels of noise, sparsity, and data scale encountered. Furthermore, the complexity of biological systems, with their intricate intercellular communications and numerous

---

*Equal contribution.
†Corresponding Author: Manolis Kellis.

38th Conference on Neural Information Processing Systems (NeurIPS 2024).

molecules involved, presents another challenge in analyzing them, especially when limited by small sample sizes, which can hinder reproducibility in biomedical research[3].

Recent advances in machine learning models especially generative models have driven rapid development in analyzing single-cell sequencing data to understand fundamental mechanisms of biology. These models achieved state-of-the-art (SOTA) performances across a range of single-cell tasks[4, 5, 6, 7, 8], including clustering analysis, batch correction, data denoising and imputation, and *in silico* data generation[3]. However, most methods focused on single-cell RNA-seq data and there is little work focusing on scATAC-seq data which is more sparse and high dimensional. Furthermore, these models treat each task with different frameworks, ignoring the potential correlations across these tasks.

To tackle these challenges, it's essential to create generative models that can not only produce high-quality scATAC-seq data but also maintain the integrity of biological representations for subsequent analysis. Recently, diffusion models [9, 10, 11] have emerged as a promising tool, demonstrating impressive results on realistic imaging generation [12, 13, 14], audio synthesis [15, 16], and molecular ligand design [17, 18]. Nevertheless, the latent variables of these diffusion models lack semantic meaning and are not interpretable since they aim to learn the diffused noise, which is not suitable for scATAC-seq data representation learning to uncover the underlying biological patterns (i.e., cellular heterogeneity within a tissue or organism). Since single-cell sequencing data is typically represented as bag-of-words, where there is no upper limit to the value of peak calling per cell and the order of peaks does not affect the cell, the diffusion model, which is designed for continuous data, is less suited for processing discrete data. To address this, we can employ a latent diffusion model[13] that utilizes a pretrained autoencoder to transform scATAC-seq samples into a continuous latent space. This transformation facilitates the effective training of the diffusion model. Additionally, it provides access to an efficient and compact latent space where high-frequency and imperceptible details are abstracted away, allowing the model to focus more on the important and variable genomic fragments.

In this study, we propose ATAC-Diff, a versatile informative latent diffusion model, to analyze and generate scATAC-seq data by introducing semantically meaningful latent space within a unified framework. Inspired by diffusion autoencoder[19] and InfoVAE[20], we introduce an auxiliary encoder equipped with Gaussian Mixture Model (GMM) as the prior of the latent space to learn the cell representation. The low dimensional cell embeddings could enhance the diffusion model to capture intrinsic high-level factors of variation present in the heterogeneous scATAC-seq data. By optimizing the mutual information between the cell embeddings and real data points, ATAC-Diff could avoid ignoring the latent variables as the conditional information when utilizing ELBO as the objectivity[21]. To the best of our knowledge, we are the first to utilize the diffusion model for scATAC-seq data generation and analysis. We conduct comprehensive experiments to demonstrate that ATAC-diff achieves SOTA or comparable performances across a range of tasks, including realistic and conditional generation, denoising, imputation, and subgroup clustering, compared to the existing models specifically designed for these tasks.

We summarize the contributions of this work as follows:

- We propose a versatile framework ATAC-Diff which can generate *in silico* scATAC-seq data with conditional information and reverse the genomic latent representation to reveal the underlying principles.

- We are the first study to harness diffusion based models for scATAC-seq data generation and analysis.

- We equip the diffusion backbone module with an auxiliary encoder which utilizes GMM as the latent feature extractor and mutual information to regulate the variational objectivities.

- We provide evidence that ATAC-Diff exhibits the capability to yield high-quality scATAC-seq data and cell latent embeddings, surpassing or achieving performance levels comparable to SOTA models.

## 2 Background

The diffusion model is formulated as two Markov chains: diffusion process and reverse process. Given a variance preserve schedule, the diffusion process transforms the real data $\mathbf{x}_0$ to a latent variable distribution $p(\mathbf{x}_{0:T})$ and finally the data is diffused into predefined Gaussian noise $x_T$

through the time step setting $t = 1...T$. The transform distribution is formulated as a fixed Markov chain which gradually adds Gaussian noise to the data with a user-defined variance schedule:

$$q\left(\mathbf{x}_t \mid \mathbf{x}_{t-1}\right) = \mathcal{N}\left(\mathbf{x}_t; \sqrt{1 - \beta_t}\mathbf{x}_{t-1}, \beta_t\mathbf{I}\right), \tag{1}$$

where $\mathbf{x}_t$ is the state of $\mathbf{x}$ at the time step $t$ which is obtained by adding Gaussian noise to the former state $\mathbf{x}_{t-1}$ and $\beta_t$ controls the degree of the noise. The distribution delineates a Gaussian distribution centered on the incrementally corrupted data state $x_t$. We could obtain a closed form from the input data $\mathbf{x}_0$ to $\mathbf{x}_T$ in a tractable way. The posterior distribution could be factorized as:

$$q\left(\mathbf{x}_{1:T} \mid \mathbf{x}_0\right) = \prod_{t=1}^{T} q(\mathbf{x}_t \mid \mathbf{x}_{t-1}) \tag{2}$$

Let $\bar{\alpha}_t = \prod_{s=1}^{t} 1 - \beta_s$, we could sample $x_t$ at any arbitrary time step $t$ in a closed form by utilizing the reparameterization trick[9]:

$$q\left(\mathbf{x}_t \mid \mathbf{x}_0\right) = \mathcal{N}\left(\mathbf{x}_t; \sqrt{\bar{\alpha}_t}\mathbf{x}_0, (1 - \bar{\alpha}_t)\mathbf{I}\right). \tag{3}$$

If the time step is sufficiently large, the final distribution will approach that of a standard Gaussian distribution. The reverse process aims to reconstruct the original data $\mathbf{x}_0$ from the diffused data $\mathbf{x}_T$, sampled from Gaussian distribution $\mathcal{N}(0, I)$, which is accomplished through the diffusion process. The reverse process can also be factorized as a Markov chain:

$$p_\theta\left(\mathbf{x}_{0:T-1} \mid \mathbf{x}_T\right) = \prod_{t-1}^{T} p\left(\mathbf{x}_{t-1} \mid \mathbf{x}_t\right) \tag{4}$$

We aim to investigate the iterative reverse process $p(\mathbf{x}_{t-1}|\mathbf{x})$. However, Estimating the distribution $p\left(\mathbf{x}_{t-1} \mid \mathbf{x}_t\right)$ is hard to estimate unless the gap between $t - 1$ and $t$ is infinitesimally small ($T = \infty$)[10]. Therefore, a learned Gaussian transitions $p_\theta\left(\mathbf{x}_{t-1} \mid \mathbf{x}_t\right)$ is devised to approximate $p\left(\mathbf{x}_{t-1} \mid \mathbf{x}_t\right)$ at every time step:

$$p_\theta\left(\mathbf{x}_{t-1} \mid \mathbf{x}_t\right) = \mathcal{N}\left(\mathbf{x}_{t-1}; \boldsymbol{\mu}_\theta\left(\mathbf{x}_t, t\right), \sigma_t^2\mathbf{I}\right) \tag{5}$$

Following previous work ([9]), $\boldsymbol{\mu}_\theta\left(\mathbf{x}_t, t\right)$ can be modeled as:

$$\boldsymbol{\mu}_\theta\left(\mathbf{x}_t, t\right) = \frac{1}{\sqrt{1 - \beta_t}}\left(\mathbf{x}_t - \frac{\beta_t}{\sqrt{1 - \bar{\alpha}_t}}\boldsymbol{\epsilon}_\theta\left(\mathbf{x}_t, t\right)\right), \tag{6}$$

where $\boldsymbol{\epsilon}_\theta$ is a neural network w.r.t trainable parameters $\theta$. Having formulated the reverse process, we could maximize the likelihood of the training data as our object. Since directly calculating the likelihood is intractable, we adopt the evidence lower bound (ELBO) [9] to optimize.

$$\mathbb{E}\left[\log p_\theta\left(\mathbf{x}\right)\right] \geq -\mathbb{E}_q[\underbrace{D_{\mathrm{KL}}\left(q\left(\mathbf{x}_T \mid \mathbf{x}_0\right) \| p\left(\mathbf{x}_T\right)\right)}_{\mathcal{L}_0} - \sum_{t=2}^{T} \underbrace{D_{\mathrm{KL}}\left(q\left(\mathbf{x}_{t-1} \mid \mathbf{x}_t, \mathbf{x}_0\right) \| p_\theta\left(\mathbf{x}_{t-1} \mid \mathbf{x}_t\right)\right)}_{\mathcal{L}_t} +$$

$$\underbrace{\log p_\theta\left(\mathbf{x}_0 \mid \mathbf{x}_1\right)}_{\mathcal{L}_T}] = \mathcal{L}_D.$$

$$\tag{7}$$

## 3  Method

In this section, we formally present our proposed model ATAC-Diff for scATAC-seq data analysis and generation. As illustrated in Figure 1, ATAC-Diff is based on the conditional latent diffusion model equipped with the auxiliary module which provides the additional latent variables. Specifically, we use fragment counts to represent the scATAC-seq data. Then we compress the scATAC-seq data (fragment counts) into lower dimensional latent space. Here, we adopt the autoencoder (AE) framework, where the encoder $\mathrm{Enc}_{\mathrm{AE}}$ encodes raw data $x_{\mathrm{raw}}$ into latent domain $x_0 = \mathrm{Enc}_{\mathrm{AE}}(x_{\mathrm{raw}})$ and the decoder $\mathrm{Dec}_{\mathrm{AE}}$ learns to decode $x_0$ back to raw data $x_{\mathrm{raw}}$. This framework can be trained by minimizing the reconstruction objective. The auxiliary encoder $z = \mathrm{Enc}_\phi(\mathbf{x}_0)$ learns to map

an input ATAC-seq data to a semantically meaningful representation $z$. By incorporating a latent variable of this nature, the diffusion model can prioritize high-level semantic information during the generation process, thereby producing data of superior quality. Additionally, this approach enables the utilization of meaningful representations at the hidden layer for downstream task analysis. Our work is inspired by the recent success of Diffusion Autoencoders[19], but learning latent variables for the sparse scATAC-seq data is however challenging. We address this by introducing GMM to extract the distinct features from the latent space and utilize mutual information to maximize the shared information between latent variables and real scATAC-seq data. We elaborate on the auxiliary module in Section 3.1 and the conditional diffusion model in Section 3.2. Finally, we briefly describe the training and sampling scheme in section 3.3.

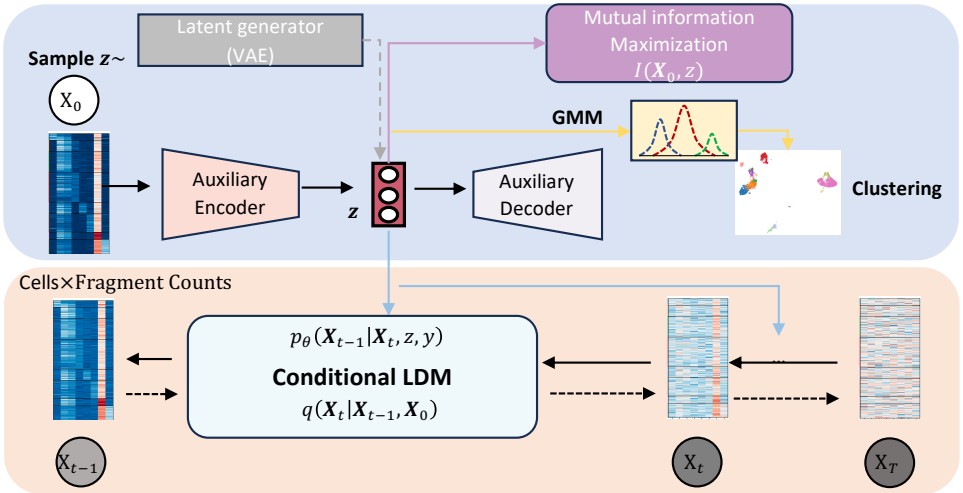

Figure 1: Overview of ATAC-Diff framework. ATAC-Diff is a versatile informative diffusion model of scATAC-seq data analysis and generation. It leverages the auxiliary model to yield low-dimensional meaningful semantic variables as the conditional information to help generate high-quality data while the reserved potential biological information could be applied for downstream tasks like cell type identification.

## 3.1 Informative auxiliary module

### 3.1.1 Semantic encoder

The goal of the informative auxiliary module is two-fold. The first is to summarize an input scATAC-seq data into a semantic representation $z = \text{Enc}_\phi(\mathbf{x}_0)$ which contains fine-grained information to assist the diffusion decoder $p_\theta(\mathbf{x}_{t-1}|\mathbf{x}_t, z)$ denoise and predict the output scATAC-seq data. The second one is that the meaningful representation yielded by the auxiliary module could be utilised for downstream analysis such as cell relationships visualization and cell heterogeneity identification.

The latent variables $z$, unlike the latent variables in the diffusion process, are flexible and can represent a low-dimensional vector of latent factors of variation which could be either continuous or discrete. In order to infer $z$, we design the approximate posterior $q_\theta(z|\mathbf{x}_0)$ which could be any architecture for the encoder. In our experiments, the encoder shares the same architecture in the diffusion backbone as the transformer encoder, which is formulated as:

$$z = \text{LayerNorm}(\mathbf{x}_0 + \text{FFL}(\text{MHA}(\mathbf{x}_0))), \tag{8}$$

where FFL is the feed-forward layer, MHA is the multi-head self-attention layer.

### 3.1.2 Semantic prior

To learn the latent distribution of the input data and enable the unconditional sampling of $\mathbf{x}_0$ in the diffusion model, we model the semantic prior $p(z)$ over data features. Usually, the prior over the latent variables is commonly an isotropic Gaussian[22]. However, utilizing a Gaussian distribution as

the prior probability distribution $p(z)$ restricts the latent representation to a single mode while the raw features of the scATAC-seq data are multi-modal with different cell types [23]. Thus, we apply GMM as the posterior to learn multi-modal latent representations[8, 24]. We predefined categorical $c \in \{1, ...K\}$ where $K$ is a predefined number of components in the mixture, e.g., the number of cell types. The joint probability $p(\mathbf{x}_0, z, c)$ could be modeled as:

$$p_\phi(\mathbf{x}_0, z, c) = p_\phi(\mathbf{x}_0|z)p_\phi(z|c)p(c), \tag{9}$$

where $p(z|c)$ is mixture of Gaussian distribution parametrized by a series of $\mu_c$ and $\sigma_c$. Then we define each factorized probability as:

$$p(c) = \text{Cat}(c|\pi), p_\phi(z|c) = \mathcal{N}(c|\mu_c, \sigma_c^2 \mathbf{I}) \tag{10}$$

We employ Kullback-Leibeler (KL) divergence to calculate the regularization term to enforce the latent variable $z$ to the GMM manifold: $\mathcal{R}_{\text{GMM}} = D_{KL}(q_\phi(\mathbf{z}, c \mid \mathbf{x_0})\|p(\mathbf{z}, c))$

### 3.1.3 Consistent latent variables with mutual information maximization as regularizer

Diffusion models can be considered as a special realisation of hierarchical VAE[25] which the encoder contains no learnable parameters. The iterative decoder model a complex decoding distribution $p_\theta(\mathbf{x}_0|\mathbf{x}_{1:T}, z)$ but suffers from the ignoring of low-dimensional latent variables $z$ [20] since high-dimensional $\mathbf{x}_0$ does not depend on $z$, leading to degrading to unconditional generation. We tackle this issue by maximizing the mutual information (MI) between $z$ and $\mathbf{x}_0$, assuming meaningful semantic variables reserve high MI with the real data point. We define MI of $\mathbf{x}_0$ and $z$ as:

$$I(\mathbf{x}_0; z) = H(z) - H(z|\mathbf{x}_0) = \mathbb{E}_{q_\phi(\mathbf{x}_0, z)}[\log \frac{q_\phi(z|\mathbf{x}_0)}{q_\phi(z)}], \tag{11}$$

where $H(z)$ and $H(z|\mathbf{x}_0)$ are the marginal information entropy and conditional information entropy, and $q_\phi(z)$ is the parameterized posterior of mixture of Gaussian distribution. Maximization of the mutual information enables the model to generate $\mathbf{x}_0$ which can infer $z$, thus avoiding the ignorance issue.

Finally, we employ an auxiliary decoder to reconstruct the latent variables to recover the real data point. The auxiliary decoder could help the auxiliary module to consistently generate semantic variables $z$ by maximizing the likelihood of $p(\mathbf{x}_0|z)$, which can be considered as complementary information to mutual information $I(\mathbf{x}_0; z)$. Furthermore, this could prevent the latent variable $z$ from generating into the pure mixture of Gaussians, which in turn interferes with the generation of $\mathbf{x}_0$. We define the reconstruction loss as:

$$\mathcal{L}_z = \mathbb{E}_{q(\mathbf{z}, c|\mathbf{x_0})}[\log \text{p}(\mathbf{x_0}|\mathbf{z})] \tag{12}$$

### 3.2 Conditional diffusion decoder

Having formulated the latent auxiliary module, we conditioned the diffusion models on the latent variables $z$ and other conditional information $y$ such as cell types, tissue, and other omics data (e.g. scRNA-seq data). Then the learning objectivity is converted to $p_\theta(\mathbf{x}_{t-1} \mid \mathbf{x}_t, z, y)$. Hence, Eq. 4 and Eq. 5 becomes the conditional format:

$$p_\theta(\mathbf{x}_{0:T-1} \mid \mathbf{x}_T) = \prod_{t-1}^{T} p(\mathbf{x}_{t-1} \mid \mathbf{x}_t, z, y) \tag{13}$$

$$p_\theta(\mathbf{x}_{t-1} \mid \mathbf{x}_t) = \mathcal{N}(\mathbf{x}_{t-1}; \boldsymbol{\mu}_\theta(\mathbf{x}_t, t, z, y), \sigma_t^2 I) \tag{14}$$

Since the diffusion model predefined the diffusion process by the user-defined variance schedules, we could encode $x_0$ to the stochastic $\mathbf{x}_t$ by running the deterministic process in Eq. 3:

$$\mathbf{x}_t = \sqrt{\bar{\alpha}_t}\mathbf{x}_0 + \sqrt{(1 - \bar{\alpha}_t)}\epsilon, \tag{15}$$

where $\epsilon$ is sampled from $\mathbf{N}(0, I)$. Similar to the auxiliary module, we also utilize transformer to model $\boldsymbol{\mu}_\theta(\mathbf{x}_t, t, z, y)$. For the conditional information $y$ embedding, we also utilize MLPs to convert the raw representation of the features to the final conditional set: $\phi(y) = \{\text{MLP}_i(y)\}$. where $i \in 1, ...L$.

Then we utilize the cross-attention mechanism to enable the prediction of $\mathbf{x}_0$ to be conditioned on the specific attribute information $y$ and $z$.

$$\text{Attention}\,(Q, K, V) \;=\; \text{softmax}\left(\frac{QK^T}{\sqrt{d}}\right)V,$$

$$Q, K, V = \mathbf{W_Q}(x_{emb}), \mathbf{W_K}(\phi(y, z)), \mathbf{W_V}(\phi(y, z)), \tag{16}$$

where $\mathbf{W_Q}, \mathbf{W_K}, \mathbf{W_V}$ are parameterized linear layers. For the categorical condition information such as cell type, we train a classifier on the latent embeddings $z$, and then select the corresponding the latent embeddings for certain cell type.

## 3.3 Training process

Having formulated the informative auxiliary and the conditional diffusion decoder, the training of the reverse process is performed by optimizing the evidence lower bound (ELBO) on negative log-likelihood (conditional form of Eq. 7) with the auxiliary module regulation. We train ATAC-Diff by the following form:

$$\mathcal{L}_{\text{ATAC-Diff}} = \mathcal{L}_D + \mathcal{L}_z + \alpha I(\mathbf{x}_0; z) + \lambda \mathcal{R}_{\text{GMM}}$$

$$= \mathbb{E}_{(\mathbf{x}_1, z, y)}[\log p_\theta\,(\mathbf{x}_0 \mid \mathbf{x}_1, z, y)] - \sum_{t=2}^{T} \mathbb{E}_{\mathbf{x}_0, \mathbf{x}_t}[D_{\text{KL}}\,(q\,(\mathbf{x}_{t-1} \mid \mathbf{x}_t, \mathbf{x}_0)\,\|\,p_\theta\,(\mathbf{x}_{t-1} \mid \mathbf{x}_t, z, y))]$$

$$+ \mathbb{E}_{q_a(\mathbf{z}, c \mid \mathbf{x_0})}[\log p_a(\mathbf{x_0}|\mathbf{z})] + (\lambda - \alpha - 1)\mathbb{E}_{q(\mathbf{x_0})}(D_{KL}(q_\phi(\mathbf{z}, c \mid \mathbf{x_0})\|p(\mathbf{z}, c))$$

$$+ \alpha D_{KL}(q_\phi(z)\|p(z)) \tag{17}$$

The full derivation can be found in the Appendix. The third term and fourth term can be considered as the ELBO objectivity of VAE model, which are feasible to calculate. However, the last term is intractable to calculate since we cannot evaluate $q_\phi(z)$. Following [20], we could sample $z \sim q_\phi(z|\mathbf{x}_0)$ which $\mathbf{x}_0 \sim q(\mathbf{x}_0)$ and then optimize it by likelihood free optimization techniques [26]. Empirically, the ELBO of diffusion models could be simplified as:

$$L_t^{\text{simple}} = \mathbb{E}_{\mathbf{x}_0, \mathbf{x}_t}\left[\|\mathbf{x}_t - \mathbf{x}_\theta(\mathbf{x}_t, t)\|^2\right]$$

$$= \mathbb{E}_{\mathbf{x}_0, \mathbf{x}_t}\left[\|\mathbf{x}_t - \mathbf{x}_\theta(\sqrt{\bar{\alpha}_t}\mathbf{x}_0 + \sqrt{1 - \bar{\alpha}_t}\mathbf{x}_t, t, z, y)\|^2\right]$$

In fact, we can directly model $\mu_\theta$ by utilizing $\mathbf{x}_\theta$ instead of $\epsilon_\theta$ since Eq.6 can be rewritten as:

$$\boldsymbol{\mu}_\theta\,(\mathbf{x}_t, t) = \frac{\sqrt{\alpha_t}(1 - \bar{\alpha}_{t-1})}{1 - \bar{\alpha}_t}\mathbf{x}_t + \frac{\sqrt{\bar{\alpha}_{t-1}}\beta_t}{1 - \bar{\alpha}_t}\mathbf{x}_\theta(\mathbf{x_t}, t, z, y) \tag{18}$$

In practice, we find that predicting $\mathbf{x}_0$ will enable the model to converge faster. We speculate that the reason for this phenomenon is that the scATAC-seq data is highly sparse with noise, predicting $\epsilon$ from $\mathbf{x}_t$ which is close to the white noise is very hard, leading to a decrease in sampling quality. For the reconstruction loss of the AE, we adopt L2 norm loss.

## 3.4 Sampling process

ATAC-Diff is different from vanilla Diffusion models which is conditioned on the latent variables. In order to sample from ATAC-Diff model, we need to design a sampling strategy to sample $z$ from the latent distribution. Since the prior distribution of latent variables is the mixture of Gaussians, it is hard to sample $z$ for unconditional generation. Therefore, we could train any arbitrary generative model to sample $z$. In this study, we employ a vanilla VAE to sample the latent variables. For data denoising and imputation, we just use the same data as the inputs for both diffusion model and the auxiliary module. In general, we could calculate the mean of the reverse Gaussian transitions $\mathbf{u}_\theta$ by Eq. 18. To sample the scATAC-seq data, we first sample the chaotic state $\mathbf{x}_T$ from $\mathcal{N}(0, I)$ or use the desired noised data as inputs. $z$ is sampled by the generative model or has the same value of $x_T$. The next less chaotic state $\mathbf{x}_{t-1}$ is generated by Eq. 5. The final state $\mathbf{x}_0$ is iteratively sampled $\mathbf{x}_{t-1}$ for $T$ times.

# 4 Experiments

In this section, we evaluate our proposed model ATAC-Diff across a range of experiments on three benchmark datasets, utilizing metrics which span generation quality, denoising and imputation effects, and latent space representation analysis. The extensive experimental results suggest that ATAC-Diff achieves higher or competitive performances compared with SOTA models which are designed for individual tasks.

## 4.1 Datasets

We adopt three datasets to benchmark our model and baseline models: Forebrain [27], Hematopoiesis [28], and PBMC10k[3]. Forebrain dataset is derived from P56 mouse forebrain. Hematopoiesis dataset includes 2,000 cells during hematopoietic differentiation through FACS. PBMC10k dataset comprises peripheral blood mononuclear cells isolated from a healthy donor, with granulocytes selectively removed through cell sorting, resulting in a cell population of approximately 10,000 cells for detailed analysis.

## 4.2 Metrics

For the latent representation analysis, we extract the latent representation of cells. We evaluate the clustering results of latent representations by Normalised Mutual Information (NMI), Adjusted Rand Index (ARI), Homogeneity score (Homo) and Average Silhouette Width(ASW). To evaluate the similarities between the generated and real cells, we adopt Spearman Correlation Coefficient (SCC) and Pearson Correlation Coefficient (PCC) for performance comparison.

## 4.3 Latent representation analysis

### 4.3.1 Experiment setting

We explore the quality of the latent representations by clustering to examine if it could be used to identify cell types. For performance comparison, we apply Highly Variable Peak (HVP) method from SCANPY [29], cisTopic[7], SCALE[8], and PeakVI[5] to obtain the corresponding dimensionality reduction features. Specifically, we leverage K-Means to obtain the clusters. We evaluate the clustering performance based on NMI, ARI, Homo, and ASW.

Table 1: Clustering performance of ATAC-Diff and baseline methods on 3 scATAC-seq datasets.

| Datasets | Forebrain | | | | Hematopoiesis | | | | PBMC10k | | | |
|---|---|---|---|---|---|---|---|---|---|---|---|---|
| Methods/Metrics | NMI | ARI | Homo | ASW | NMI | ARI | Homo | ASW | NMI | ARI | Homo | ASW |
| HVP | 0.247 | 0.093 | 0.178 | -0.563 | 0.083 | 0.023 | 0.065 | -0.324 | 0.522 | 0.400 | 0.454 | -0.672 |
| PCA | 0.359 | 0.182 | 0.329 | 0.081 | 0.041 | 0.023 | 0.034 | 0.171 | 0.626 | 0.419 | 0.679 | 0.393 |
| cisTopic | 0.665 | 0.540 | 0.650 | 0.467 | 0.639 | 0.457 | 0.660 | 0.272 | 0.736 | 0.499 | 0.802 | 0.467 |
| SCALE | 0.718 | 0.657 | 0.722 | 0.515 | 0.608 | 0.404 | 0.631 | 0.371 | 0.675 | 0.451 | 0.739 | 0.434 |
| PeakVI | 0.499 | 0.377 | 0.511 | 0.372 | 0.593 | 0.398 | 0.622 | 0.384 | 0.699 | 0.458 | 0.766 | 0.433 |
| ATAC-Diff | 0.740 | 0.674 | 0.673 | 0.533 | 0.647 | 0.492 | 0.666 | 0.423 | 0.733 | 0.506 | 0.800 | 0.386 |

### 4.3.2 Experimental results

The clustering results of ATAC-Diff and the baseline models on 3 benchmark datasets are presented in Table 8. The first, second and third highest values are colored by red, green and purple. The higher the value if the metrics, the better the clustering performance performance. Among the 3 datasets based on the four metrics, ATAC-Diff yields 8 best scores and comparable scores compared with the baseline models. For instance, ATAC-Diff outperforms all the baseline models across all the metrics on Hematopoiesis dataset, defeats SOTA models by at least 1.25% in NMI, 7.66% in ARI, 0.91% in Homo, and 10.16% in ASW. Furthermore, we observe that ATAC-Diff and SCALE achieve similar performance on Forebrain dataset but ATAC-Diff surpasses it and achieves competitive performances on PBMC10k dataset.

---

[3]Downloaded PBMC10k dataset from
`https://support.10xgenomics.com/single-cell-multiome-atac-gex/datasets/1.0.0/pbmc_granulocyte_sorted_10k`, generated from a healthy donor after removing granulocytes through cell sorting

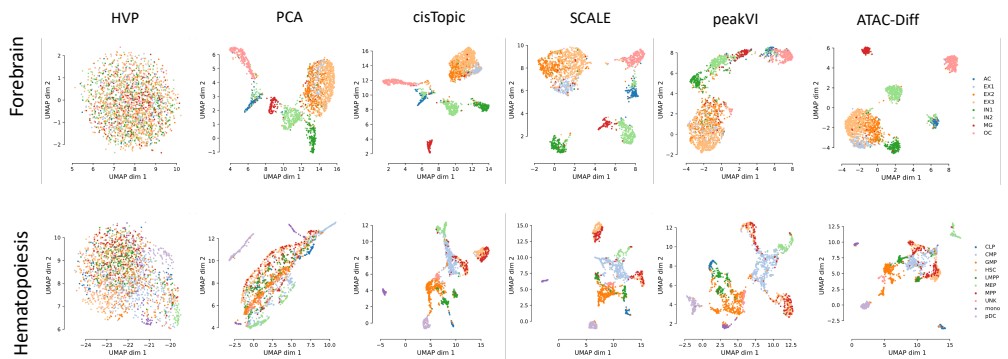

Figure 2: UMAP visualization of the highly variable peak values and extracted features from PCA, cisTopic, SCALE, PeakVI, and ATAC-Diff on Forebrain and Hematopiesis datasets. The visualization of PBMC10k is included in the Appendix.

To determine whether the methods effectively separate the cell types in the latent space, we employ UMAP to visualize the extracted features (Figure 2). Our analysis reveals that ATAC-Diff exhibits superior performance in distinguishing the cell types, whereas HVP fails to differentiate between the cell types. Furthermore, some of the other methods show overlapping results for certain cell types. In addition, our findings demonstrate that ATAC-Diff has the ability to unveil the distance between distinct cell subpopulations and their developmental trajectories. For instance, we observed that the three excitatory neuron cell types (EC1, EC2, EC3) from the Forebrain dataset exhibit close proximity to each other in the latent space of ATAC-Diff. Conversely, the distances between different cell types are significantly larger, indicating distinct cellular identities. In the Hematopoiesis dataset, we observed that the Multipotent Progenitor Cells (MPPs), which are downstream progenitor cells derived from Hematopoietic Stem Cells (HSCs), exhibit close proximity to each other in the latent space of ATAC-Diff. This finding aligns with their known differentiation relationship, further supporting the accuracy and biological relevance of our analysis.

## 4.4 Generation quality measurement

### 4.4.1 Experimental settings

In this section, we have devised two strategies to evaluate the generation capabilities of ATAC-Diff. The first strategy involves unconditional generation, where we assess whether the scATAC-seq data generated by ATAC-Diff exhibits realistic characteristics. If our model excels in capturing the data distribution, it can be leveraged to create simulated data for augmentation purposes rather than sequencing additional cells, thus conserving time and resources. Furthermore, visualizing the synthetic data generated can enhance researchers' comprehension of the distribution patterns and fundamental structure of scATAC-seq data. In this case, we anticipate that ATAC-Diff will be able to generate cells that possess the specific subpopulation features associated with the provided conditional information. Since we are the first to generate scATAC-seq data from scratch, there are no baseline models for comparison. Therefore, we have made modifications to SCALE and PeakVI, which are VAE-based models, to enable unconditional generation by utilizing random noise as inputs. For conditional generation, we have replaced the VAE module of SCALE with a conditional VAE module[4]. For the unconditional generation, we average all single cells as the ground truth. For the conditional generation, we average all single cells of the same biological cell type as the ground truth. To address the requirement of ATAC-Diff for latent variables as auxiliary information, we employ a vanilla VAE to sample the latent variables. We evaluate the generation performance based on the SCC and PCC between the mean values of generated samples and the two ground truth data.

---

[4]Unfortunately, due to the high integration of PeakVI into the scvi-tools package, we were unable to make modifications to it.

#### 4.4.2 Experimental results

We report the results of unconditional generation in Table 6. We observe that ATAC-Diff outperforms other methods in generating scATAC-seq data from scratch, yielding more realistic results. Specifically, ATAC-Diff achieves the highest SCC of 0.892 and PCC of 0.969 on Forebrain dataset, SCC of 0.822 and PCC of 0.973 on Hematopoiesis dataset, and PCC of 0.983 on PBMC10k dataset. Table 6 also displays the mean correlation values of each cell type in the context of conditional generation. We observe that ATAC-Diff outperforms conditional SCALE among all the metrics. In addition, the performance of ATAC-Diff without conditional information degraded, indicating that ATAC-Diff can fuse the conditional cell type information to generate high-quality scATAC-seq data of specific cell types. Overall, we prove that ATAC-Diff is capable of generating realistic scATAC-seq data.

Table 2: Unconditional and conditional generation performance of ATAC-Diff and other baseline methods on 3 scATAC-seq datasets.

| Datasets | Forebrain | | Hematopoiesis | | PBMC10k | |
|---|---|---|---|---|---|---|
| Methods/Metrics | SCC | PCC | SCC | PCC | SCC | PCC |
| **Unconditional Generation** | | | | | | |
| SCALE | 0.548 | 0.728 | 0.799 | 0.719 | 0.892 | 0.451 |
| PeakVI | 0.330 | 0.366 | 0.759 | 0.762 | 0.824 | 0.860 |
| ATAC-Diff | 0.925 | 0.992 | 0.927 | 0.976 | 0.964 | 0.997 |
| **Conditional Generation** | | | | | | |
| SCALE | 0.576 | 0.746 | 0.708 | 0.816 | 0.838 | 0.819 |
| ATAC-Diff w.o con | 0.418 | 0.728 | 0.496 | 0.840 | 0.828 | 0.915 |
| ATAC-Diff | 0.688 | 0.770 | 0.850 | 0.923 | 0.846 | 0.922 |

### 4.5 Data denoising and imputation

#### 4.5.1 Experimental settings

The scATAC-seq data always contains both noised and a large number of missing values due to dropout events[30]. We design two approaches to test the ability of the method for denoising and recovering missing values (imputation) to address the issues under real scenarios. For scATAC-seq data denoising, analyzing real datasets poses a challenge due to the lack of ground truth data. Following previous work[8], we mitigate this challenge to average all single cells within the same cell type, resulting in a meta-cell that serves as a good approximation of those individual cells. For scATAC-seq data imputation, previous works chose to create a corrupted matrix by randomly dropping out some non-zero entries. In our case, we follow the setting used in DCA [31] to simulate dropout events by masking 10% of the non-zero counts and setting them to zero. The probability of masking a particular non-zero count follows an exponential distribution which peaks with lower expression levels have a higher likelihood of dropout compared to genes with higher expression levels[32]. For comparison, we choose SCALE and peakVI as baseline models and SCC and PCC as metrics.

#### 4.5.2 Experimental Results

The results of scATAC-seq data denoising and imputation of ATAC-Diff and baseline models are presented in Table 7 where the best result is highlighted in bold. For the data denoising evaluation, ATAC-Diff achieves the highest correlation of the denoised single cells with the corresponding meta-cells of each cell type on PBMC10k dataset and competitive results on the other two datasets. Although the other generative models PeakVI and SCALE perform well on some of the datasets, their performances are not robust. One potential reason is that the auxiliary latent variables enable ATAC-Diff to capture the high-level semantics information. Similarly, ATAC-Diff performs stable on scATAC-seq data imputation, where the generated data is closely related to the meta-cells.

## 5 Conclusion

In this work, we propose a versatile framework ATAC-Diff which is based on a diffusion model conditioned on the latent variables for scATAC-seq data analysis and generation. ATAC-Diff could utilize the proposed auxiliary module to yield the latent variables which contain high-level meaningful

Table 3: Denoising and imputation performance of ATAC-Diff and other baseline methods on 3 scATAC-seq datasets.

| Datasets | Forebrain | | Hematopoiesis | | PBMC10k | |
|---|---|---|---|---|---|---|
| Methods/Metrics | SCC | PCC | SCC | PCC | SCC | PCC |
| **Denoising** | | | | | | |
| SCALE | 0.777 | 0.870 | 0.676 | 0.726 | 0.858 | 0.945 |
| PeakVI | 0.710 | 0.760 | 0.874 | 0.879 | 0.860 | 0.948 |
| ATAC-Diff | 0.718 | 0.873 | 0.840 | 0.875 | 0.863 | 0.950 |
| **Imputation** | | | | | | |
| SCALE | 0.760 | 0.860 | 0.888 | 0.947 | 0.858 | 0.924 |
| PeakVI | 0.708 | 0.755 | 0.870 | 0.870 | 0.916 | 0.947 |
| ATAC-Diff | 0.716 | 0.861 | 0.892 | 0.909 | 0.860 | 0.949 |

information; it could be incorporated to assist the data sampling during diffusion. Additionally, these variables could be examined for downstream analyses such as cell type annotation. Based on the formulation of the mutual information, we prevent the model from ignoring the low-dimensional latent variables. Furthermore, we employ a Gaussian Mixture Model (GMM) as the latent prior and auxiliary decoder to enhance the recovery of real data, allowing the latent variables to learn the genomic semantics.

We conducted extensive experiments to evaluate the performance of ATAC-Diff on three datasets and compared it with SOTA models, indicating that the ATAC-Diff model can generate high-quality data *in silico* scATAC-seq data while effectively disentangling the cell embeddings. An intriguing potential direction for ATAC-Diff involves exploring various conditional generation scenarios for genomic discovery, such as identifying the connections between scRNA-seq data and perturbation prediction. We anticipate that ATAC-Diff can contribute to the advancement of genomic analysis in the near future of personalized medicine at the single-cell level.

# 6   Acknowledgment

The work was supported in part by NIH grants R01 AG081017-01, R01 AG067151, and HG012009-01. The work was partially supported by the grant from the Research Grants Council of the Hong Kong Special Administrative Region [CityU 11203723]. The work described in this paper was partially supported by the grants from City University of Hong Kong (2021SIRG036, CityU 9667265, CityU 11203221) and Innovation and Technology Commission (ITB/FBL/9037/22/S).

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

# A Proof of the diffusion model

We provide proofs for the derivation of several properties in ATAC-Diff.

## A.1 Marginal distribution of the diffusion process

In the diffusion process, we have the marginal distribution of the data at any arbitrary time step $t$ in a closed form:$q\left(\mathbf{x}_t \mid \mathbf{x}_0\right) = \mathcal{N}\left(\mathbf{x}_t; \sqrt{\bar{\alpha}_t}\mathbf{x}_0, \left(1 - \bar{\alpha}_t\right)\mathbf{I}\right).$

**Proof**: recall the posterior $q\left(\mathbf{x}_t \mid \mathbf{x}_{t-1}\right)$ in Eq. 3, we can obtain $\mathbf{x}_t$ using the reparameterization trick. A property of the Gaussian distribution is that if we add $\mathcal{N}(\mathbf{0}, \sigma_1^2\mathbf{I})$ and $\mathcal{N}(\mathbf{0}, \sigma_2^2\mathbf{I})$, the new distribution is $\mathcal{N}(\mathbf{0}, (\sigma_1^2 + \sigma_2^2)\mathbf{I})$

$$
\begin{aligned}
\mathbf{x}_t &= \sqrt{\alpha_t}\mathbf{x}_{t-1} + \sqrt{1 - \alpha_t}\epsilon_{t-1} \\
&= \sqrt{\alpha_t\alpha_{t-1}}\mathbf{x}_{t-2} + \sqrt{\alpha_t(1 - \alpha_{t-1})}\epsilon_{t-2} + \sqrt{1 - \alpha_t}\epsilon_{t-1} \\
&= \sqrt{\alpha_t\alpha_{t-1}}\mathbf{x}_{t-2} + \sqrt{1 - \alpha_t\alpha_{t-1}}\bar{\epsilon}_{t-2} \\
&= \ldots \\
&= \sqrt{\bar{\alpha}_t}\mathbf{x}_0 + \sqrt{1 - \bar{\alpha}_t}\bar{\epsilon},
\end{aligned}
\tag{19}
$$

where $\alpha_t = 1 - \beta_t$, $\epsilon$ and $\hat{\epsilon}$ are sampled from independent standard Gaussian distributions. Then we could derive Eq. 3

## A.2 The derivation of parameterized mean $\mu_\theta$

A learned Gaussian transitions $p_\theta\left(\mathbf{x}_{t-1} \mid \mathbf{x}_t\right)$ is devised to approximate the $q\left(\mathbf{x}_{t-1} \mid \mathbf{x}_t\right)$ of every time step: $p_\theta\left(\mathbf{x}_{t-1} \mid \mathbf{x}_t\right) = \mathcal{N}\left(\mathbf{x}_{t-1}; \boldsymbol{\mu}_\theta\left(\mathbf{x}_t, t\right), \sigma_t^2 I\right)$. $\boldsymbol{\mu}_\theta$ is parameterized as follows:

$$
\boldsymbol{\mu}_\theta\left(\mathbf{x}_t, t\right) = \frac{1}{\sqrt{\alpha_t}}\left(\mathbf{x}_t - \frac{\beta_t}{\sqrt{1 - \bar{\alpha}_t}}\boldsymbol{\epsilon}_\theta\left(\mathbf{x}_t, t\right)\right).
$$

**Proof**: the distribution $q\left(\mathbf{x}_{t-1} \mid \mathbf{x}_t\right)$ can be expanded by Bayes' rule:

$$
\begin{aligned}
&q\left(\mathbf{x}_{t-1} \mid \mathbf{x}_t\right) \\
&= q\left(\mathbf{x}_{t-1} \mid \mathbf{x}_t, \mathbf{x}_0,\right) \\
&= q\left(\mathbf{x}_t \mid \mathbf{x}_{t-1}, \mathbf{x}_0\right)\frac{q\left(\mathbf{x}_{t-1} \mid \mathbf{x}_0\right)}{q\left(\mathbf{x}_t \mid \mathbf{x}_0\right)} \\
&= q\left(\mathbf{x}_t \mid \mathbf{x}_{t-1}\right)\frac{q\left(\mathbf{x}_{t-1}\mathbf{x}_0\right)}{q\left(\mathbf{x}_t \mid \mathbf{x}_0\right)} \\
&\propto \exp\left(-\frac{1}{2}\left(\frac{\left(\mathbf{x}_t - \sqrt{\alpha_t}\mathbf{x}_{t-1}\right)^2}{\beta_t} + \frac{\left(\mathbf{x}_{t-1} - \sqrt{\bar{\alpha}_{t-1}}\mathbf{x}_0\right)^2}{1 - \bar{\alpha}_{t-1}} - \frac{\left(\mathbf{x}_t - \sqrt{\bar{\alpha}_t}\mathbf{x}_0\right)^2}{1 - \bar{\alpha}_t}\right)\right) \\
&= \exp\left(-\frac{1}{2}\left(\left(\frac{\alpha_t}{\beta_t} + \frac{1}{1 - \bar{\alpha}_{t-1}}\right)\mathbf{x}_{t-1}^2 - \left(\frac{2\sqrt{\alpha_t}}{\beta_t}\mathbf{x}_t + \frac{2\sqrt{\bar{\alpha}_{t-1}}}{1 - \bar{\alpha}_{t-1}}\mathbf{x}_0\right)\mathbf{x}_{t-1} + C\left(\mathbf{x}_t, \mathbf{x}_0\right)\right)\right) \\
&\propto \exp(-\mathbf{x}_{t-1}^2 + (\frac{\sqrt{\alpha_t}\left(1 - \bar{\alpha}_{t-1}\right)}{1 - \bar{\alpha}_t}\mathbf{x}_t + \frac{\sqrt{\bar{\alpha}_{t-1}}\beta_t}{1 - \bar{\alpha}_t}\mathbf{x}_0)\mathbf{x}_{t-1}),
\end{aligned}
\tag{20}
$$

where $C\left(\mathbf{x}_t, \mathbf{x}_0\right)$ is a constant. We can find that $q\left(\mathbf{x}_{t-1} \mid \mathbf{x}_t\right)$ is also a Gaussian distribution. We assume that:

$$
q\left(\mathbf{x}_{t-1} \mid \mathbf{x}_t, \mathbf{x}_0\right) = \mathcal{N}\left(\mathbf{x}_{t-1}; \tilde{\boldsymbol{\mu}}\left(\mathbf{x}_t, \mathbf{x}_0\right), \tilde{\beta}_t I\right),
\tag{21}
$$

where $\tilde{\beta}_t = 1/\left(\frac{\alpha_t}{\beta_t} + \frac{1}{1 - \bar{\alpha}_{t-1}}\right) = \frac{1 - \bar{\alpha}_{t-1}}{1 - \bar{\alpha}_t} \cdot \beta_t$

and $\tilde{\boldsymbol{\mu}}_t\left(\mathbf{x}_t, \mathbf{x}_0\right) = \left(\frac{\sqrt{\alpha_t}}{\beta_t}\mathbf{x}_t + \frac{\sqrt{\bar{\alpha}_{t-1}}}{1 - \bar{\alpha}_{t-1}}\mathbf{x}_0\right)/\left(\frac{\alpha_t}{\beta_t} + \frac{1}{1 - \bar{\alpha}_{t-1}}\right) = \frac{\sqrt{\alpha_t}\left(1 - \bar{\alpha}_{t-1}\right)}{1 - \bar{\alpha}_t}\mathbf{x}_t + \frac{\sqrt{\bar{\alpha}_{t-1}}\beta_t}{1 - \bar{\alpha}_t}\mathbf{x}_0.$

Then we could parameterize $\boldsymbol{\mu}_\theta(\mathbf{x}_t, t) = \frac{\sqrt{\alpha_t}(1-\bar{\alpha}_{t-1})}{1-\bar{\alpha}_t}\mathbf{x}_t + \frac{\sqrt{\bar{\alpha}_{t-1}}\beta_t}{1-\bar{\alpha}_t}\mathbf{x}_\theta$, which is presentned in Eq. 18. From Eq. 19, we have $\mathbf{x}_t == \sqrt{\bar{\alpha}_t}\mathbf{x}_0 + \sqrt{1-\bar{\alpha}_t}\epsilon$. We take this into $\tilde{\boldsymbol{\mu}}$:

$$
\begin{aligned}
\tilde{\boldsymbol{\mu}}_t &= \frac{\sqrt{\alpha_t}(1-\bar{\alpha}_{t-1})}{1-\bar{\alpha}_t}\mathbf{x}_t + \frac{\sqrt{\bar{\alpha}_{t-1}}\beta_t}{1-\bar{\alpha}_t}\frac{1}{\sqrt{\bar{\alpha}_t}}\left(\mathbf{x}_t - \sqrt{1-\bar{\alpha}_t}\epsilon_t\right) \\
&= \frac{1}{\sqrt{\alpha_t}}\left(\mathbf{x}_t - \frac{\beta_t}{\sqrt{1-\bar{\alpha}_t}}\epsilon_t\right)
\end{aligned}
$$

$\boldsymbol{\mu}_\theta$ is designed to model $\tilde{\boldsymbol{\mu}}$. Therefore, $\boldsymbol{\mu}_\theta$ has the same formulation as $\tilde{\boldsymbol{\mu}}$ but parameterizes $\epsilon$:
$\boldsymbol{\mu}_\theta(\mathbf{x}_t, t) = \frac{1}{\sqrt{\alpha_t}}\left(\mathbf{x}_t - \frac{\beta_t}{\sqrt{1-\bar{\alpha}_t}}\epsilon_\theta(\mathbf{x}_t, t)\right)$.

### A.3 The derivation of loss function

It is hard to directly calculate the conditional log-likelihood of the data. Instead, we can derive its ELBO objective for optimizing. For simplicity, the $c$ in GMM is omitted

$$
\begin{aligned}
\mathbb{E}\left[\log p_\theta(\mathbf{x}_0)\right] &= \mathbb{E}_{q(\mathbf{x}_0)}\log\mathbb{E}_{q(z)}\int p_\theta(\mathbf{x}_{0:T}|z)\,d\mathbf{x}_{1:T} \\
&= \mathbb{E}_{q(\mathbf{x}_0)}\log\int p_\theta(\mathbf{x}_{0:T}, z)\,d\mathbf{x}_{1:T}dz \\
&= \mathbb{E}_{q(\mathbf{x}_0)}\log\left(\int q(\mathbf{x}_{1:T}\mid\mathbf{x}_0)q_\phi(z|x_0)\frac{p_\theta(\mathbf{x}_{0:T}, z)}{q(\mathbf{x}_{1:T}\mid\mathbf{x}_0)q_\phi(z|x_0)}\,d\mathbf{x}_{1:T}dz\right) \quad (22) \\
&= \mathbb{E}_{q(\mathbf{x}_0)}\log\left(\mathbb{E}_{q(\mathbf{x}_{1:T}|\mathbf{x}_0)}\mathbb{E}_{q_\phi(z|x_0)}\frac{p_\theta(\mathbf{x}_{0:T}, z)}{q(\mathbf{x}_{1:T}\mid\mathbf{x}_0))q_\phi(z|x_0)}\right) \\
&\geq \mathbb{E}_{q(\mathbf{x}_{0:T})}\log\mathbb{E}_{q_\phi(z|x_0)}\frac{p_\theta(\mathbf{x}_{0:T}, z)}{q(\mathbf{x}_{1:T}\mid\mathbf{x}_0)q_\phi(z|x_0))}
\end{aligned}
$$

Then we further derive the conditional ELBO objective:

$$\mathbb{E}_{q(\mathbf{x}_{0:T})}\mathbb{E}_{q_\phi(z|\mathbf{x}_0)}\left[\log\frac{p_\theta\left(\mathbf{x}_{0:T},z\right)}{q\left(\mathbf{x}_{1:T}\mid\mathbf{x}_0\right)q_\phi(z|\mathbf{x}_0)}\right]$$

$$=\mathbb{E}_q\left[\log\frac{p_\theta\left(\mathbf{x}_T\right)p_\theta(z)\prod_{t=1}^T p_\theta\left(\mathbf{x}_{t-1}\mid\mathbf{x}_t,z\right)}{q_\phi(z|\mathbf{x}_0)\prod_{t=1}^T q\left(\mathbf{x}_t\mid\mathbf{x}_{t-1}\right)}\right]$$

$$=\mathbb{E}_q\left[\log\frac{p_\theta\left(\mathbf{x}_T\right)p_\theta(z)}{q_\phi(z|\mathbf{x}_0)}+\sum_{t=1}^T\log\frac{p_\theta\left(\mathbf{x}_{t-1}\mid\mathbf{x}_t,z\right)}{q\left(\mathbf{x}_t\mid\mathbf{x}_{t-1}\right)}\right]$$

$$=\mathbb{E}_q\left[\log\frac{p_\theta\left(\mathbf{x}_T\right)p_\theta(z)}{q_\phi(z|\mathbf{x}_0)}+\sum_{t=2}^T\log\frac{p_\theta\left(\mathbf{x}_{t-1}\mid\mathbf{x}_t,z\right)}{q\left(\mathbf{x}_t\mid\mathbf{x}_{t-1}\right)}+\log\frac{p_\theta\left(\mathbf{x}_0\mid\mathbf{x}_1,z\right)}{q\left(\mathbf{x}_1\mid\mathbf{x}_0\right)}\right]$$

$$=\mathbb{E}_q\left[\log\frac{p_\theta\left(\mathbf{x}_T\right)p_\theta(z)}{q_\phi(z|\mathbf{x}_0)}+\sum_{t=2}^T\log\left(\frac{p_\theta\left(\mathbf{x}_{t-1}\mid\mathbf{x}_t,z\right)}{q\left(\mathbf{x}_{t-1}\mid\mathbf{x}_t,\mathbf{x}_0\right)}\cdot\frac{q\left(\mathbf{x}_{t-1}\mid\mathbf{x}_0\right)}{q\left(\mathbf{x}_t\mid\mathbf{x}_0\right)}\right)+\log\frac{p_\theta\left(\mathbf{x}_0\mid\mathbf{x}_1,z\right)}{q\left(\mathbf{x}_1\mid\mathbf{x}_0\right)}\right]$$

$$=\mathbb{E}_q[\log\frac{p_\theta\left(\mathbf{x}_T\right)p_\theta(z)}{q_\phi(z|\mathbf{x}_0)}+\sum_{t=2}^T\log\frac{p_\theta\left(\mathbf{x}_{t-1}\mid\mathbf{x}_t,z\right)}{q\left(\mathbf{x}_{t-1}\mid\mathbf{x}_t,\mathbf{x}_0\right)}+\sum_{t=2}^T\log\frac{q\left(\mathbf{x}_{t-1}\mid\mathbf{x}_0\right)}{q\left(\mathbf{x}_t\mid\mathbf{x}_0\right)}+$$

$$\log\frac{p_\theta\left(\mathbf{x}_0\mid\mathbf{x}_1,z\right)}{q\left(\mathbf{x}_1\mid\mathbf{x}_0\right)}]$$

$$=\mathbb{E}_q\left[\log\frac{p_\theta\left(\mathbf{x}_T\right)p_\theta(z)}{q_\phi(z|\mathbf{x}_0)}+\sum_{t=2}^T\log\frac{p_\theta\left(\mathbf{x}_{t-1}\mid\mathbf{x}_t,z\right)}{q\left(\mathbf{x}_{t-1}\mid\mathbf{x}_t,\mathbf{x}_0\right)}+\log\frac{q\left(\mathbf{x}_1\mid\mathbf{x}_0\right)}{q\left(\mathbf{x}_T\mid\mathbf{x}_0\right)}+\log\frac{p_\theta\left(\mathbf{x}_0\mid\mathbf{x}_1,z\right)}{q\left(\mathbf{x}_1\mid\mathbf{x}_0\right)}\right]$$

$$=\mathbb{E}_q\left[\log\frac{p_\theta\left(\mathbf{x}_T|z\right)}{q\left(\mathbf{x}_T\mid\mathbf{x}_0\right)}+\sum_{t=2}^T\log\frac{p_\theta\left(\mathbf{x}_{t-1}\mid\mathbf{x}_t,z\right)}{q\left(\mathbf{x}_{t-1}\mid\mathbf{x}_t,\mathbf{x}_0\right)}+\log p_\theta\left(\mathbf{x}_0\mid\mathbf{x}_1,z\right)+\log\frac{p_\theta(z)}{q_\phi(z|\mathbf{x}_0)}\right]$$

$$=\mathbb{E}_q\left[\underbrace{-D_{\mathrm{KL}}\left(q\left(\mathbf{x}_T\mid\mathbf{x}_0\right)\|p_\theta\left(\mathbf{x}_T|z\right)\right)}_{L_T}-\right.$$

$$\left.\sum_{t=2}^T\underbrace{D_{\mathrm{KL}}\left(q\left(\mathbf{x}_{t-1}\mid\mathbf{x}_t,\mathbf{x}_0\right)\|p_\theta\left(\mathbf{x}_{t-1}\mid\mathbf{x}_t,z\right)\right)}_{L_t}+\underbrace{\log p_\theta\left(\mathbf{x}_0\mid\mathbf{x}_1,z\right)}_{L_0}+\log\frac{p_\theta(z)}{q_\phi(z|\mathbf{x}_0)}\right]$$

$$(23)$$

### A.4 Proof of the ATAC-Diff Objectivity

For simplicity, we utilize the last term $\log\frac{p_\theta(z)}{q_\phi(z|\mathbf{x}_0)}$ and the regular term of GMM and mutual information to obtain the final form of the optimizing objectivity in Eq. 7 (We also omit c in GMM for simplicity).

$$\mathbb{E}_q\left[\log\frac{p_\theta(z)}{q_\phi(z|\mathbf{x}_0)}+\alpha I(\mathbf{x}_0;z)+\lambda\mathcal{R}_{\mathrm{GMM}}\right]$$

$$=\mathbb{E}_q\left[\log\frac{p_\theta(z)}{q_\phi(z|\mathbf{x}_0)}+\alpha\log\frac{q_\phi(z)}{q_\phi(z|\mathbf{x}_0)}+\lambda\log\frac{q_\phi(z|\mathbf{x}_0)}{p(z)}\right]$$

$$=\mathbb{E}_q\left[\log\frac{q_\phi(z|\mathbf{x}_0)^{\lambda-\alpha-1}q_\phi(z)^\alpha}{p(z)^{\lambda-1}}\right]$$

$$=\mathbb{E}_q\left[\log\frac{q_\phi(z|\mathbf{x}_0)^{\lambda-\alpha-1}q_\phi(z)^\alpha}{p(z)^{\lambda-\alpha-1}p(z)^\alpha}\right]$$

$$=\mathbb{E}_q\left[\log\frac{q_\phi(z|\mathbf{x}_0)^{\lambda-\alpha-1}}{p(z)^{\lambda-\alpha-1}}+\log\frac{q_\phi(z)^\alpha}{p(z)^\alpha}\right]$$

$$=(\lambda-\alpha-1)D_{KL}(q_\phi(z|\mathbf{x}_0)\|p(z))+\alpha D_{KL}(q_\phi(z)\|p(z))$$

$$(24)$$

Finally, we could derive the objectivity as:

$$
\begin{aligned}
&\mathcal{L}_{\text{ATAC-Diff}} \\
&= \mathbb{E}_{(\mathbf{x}_1, z, y)}[\log p_\theta(\mathbf{x}_0 \mid \mathbf{x}_1, z, y)] \\
&- \sum_{t=2}^{T} \mathbb{E}_{\mathbf{x}_0, \mathbf{x}_t}[D_{\text{KL}}(q(\mathbf{x}_{t-1} \mid \mathbf{x}_t, \mathbf{x}_0) \| p_\theta(\mathbf{x}_{t-1} \mid \mathbf{x}_t, z, y)))] \\
&+ \mathbb{E}_{q_a(\mathbf{z}, c \mid \mathbf{x}_0)}[\log p_a(\mathbf{x}_0 \mid \mathbf{z})] \\
&+ (\lambda - \alpha - 1)\mathbb{E}_{q(\mathbf{x}_0)}(D_{KL}(q_\phi(\mathbf{z}, c \mid \mathbf{x}_0) \| p(\mathbf{z}, c)) \\
&+ \alpha D_{KL}(q_\phi(z) \| p(z))
\end{aligned}
\tag{25}
$$

# B Experiments details

## B.1 Dataset and implementation

We summarize the statistic information of all processed datasets in Table 4.

| Dataset | #cells | #peaks | #cell types | Reference |
|---------|--------|--------|-------------|-----------|
| Forebrain | 2088 | 11285 | 8 | Preissl et al.,2018 |
| Hematopoiesi | 2034 | 103151 | 10 | Buenrostro et al., 2018 |
| PBMC10k | 9631 | 107194 | 19 | 10xgenomics, 2020 |

Table 4: The statistic summary of datasets

For all three datastes, ATAC-Diff is trained by AdamW [33] optimizer for 10K iterations with a batch size of 256 and a learning rate of 0.0001 on one NVIDIA V100 GPU card. We set $\alpha$ as 0.1 and $\lambda$ as 1.1001.

## B.2 Ablation study

In this study, we have removed the GMM and MI modules to conduct the ablation study for each task.

Table 5: Ablastion study for clustering.

| Datasets | Forebrain | | | | Hematopoiesis | | | | PBMC10k | | | |
|----------|------|------|------|------|------|------|------|------|------|------|------|------|
| Methods/Metrics | NMI | ARI | Homo | ASW | NMI | ARI | Homo | ASW | NMI | ARI | Homo | ASW |
| ATAC w.o GMM | 0.558 | 0.438 | 0.556 | 0.202 | 0.489 | 0.297 | 0.505 | 0.222 | 0.586 | 0.288 | 0.655 | 0.137 |
| ATAC w.o MI | 0.596 | 0.451 | 0.602 | 0.120 | 0.490 | 0.299 | 0.510 | 0.077 | 0.590 | 0.231 | 0.657 | 0.032 |

Table 6: Ablation study for unconditional and conditional generation.

| Datasets | Forebrain | | Hematopoiesis | | PBMC10k | |
|----------|------|------|------|------|------|------|
| Methods/Metrics | SCC | PCC | SCC | PCC | SCC | PCC |
| **Unconditional Generation** | | | | | | |
| ATAC w.o GMM | 0.919 | 0.991 | 0.886 | 0.949 | 0.693 | 0.710 |
| ATAC w.o MI | 0.916 | 0.991 | 0.904 | 0.962 | 0.704 | 0.729 |
| **Conditional Generation** | | | | | | |
| ATAC w.o GMM | 0.678 | 0.768 | 0.845 | 0.910 | 0.823 | 0.911 |
| ATAC w.o MI 0.681 | 0.769 | 0.848 | 0.913 | 0.831 | 0.920 | |

## B.3 Euclidean distances of different cell types

We present a similarity matrix of the latent embedding based on Euclidean distances. Specifically, we average the latent embeddings within each cell type population and calculate the Euclidean distances across cell types.

Table 7: Ablation study for denoising and imputation.

| Datasets | Forebrain | | Hematopoiesis | | PBMC10k | |
|---|---|---|---|---|---|---|
| Methods/Metrics | SCC | PCC | SCC | PCC | SCC | PCC |
| **Denoising** | | | | | | |
| ATAC w.o GMM | 0.701 | 0.867 | 0.823 | 0.851 | 0.843 | 0.932 |
| ATAC w.o MI 0.706 | 0.868 | 0.831 | 0.856 | 0.851 | 0.940 | |
| **Imputation** | | | | | | |
| ATAC w.o GMM | 0.710 | 0.841 | 0.887 | 0.898 | 0.833 | 0.935 |
| ATAC w.o MI 0.712 | 0.845 | 0.887 | 0.901 | 0.831 | 0.931 | |

Our results show that ATAC-Diff can effectively capture the relationships between distinct cell subpopulations and their developmental trajectories. For example, in the Forebrain dataset, the three excitatory neuron subtypes (EC1, EC2, EC3), which are biologically similar, cluster closely in ATAC-Diff's latent space based on the correlation of their embeddings. In contrast, more biologically distant cell types display greater separation. Similarly, in the Hematopoiesis dataset, Multipotent Progenitor Cells (MPPs) and Hematopoietic Stem Cells (HSCs) exhibit proximity, reflecting their known differentiation path and reinforcing the biological relevance of our findings.

Table 8: The Euclidean distances across different cell types.

| Comparison | AC | EX1 | EX2 | EX3 | IN1 | IN2 | MG | OC |
|---|---|---|---|---|---|---|---|---|
| AC | - | 0.8307 | 0.7431 | 0.7594 | 0.8508 | 0.7237 | 0.8550 | 0.8520 |
| EX1 | 0.8307 | - | 0.4773 | 0.4667 | 0.6548 | 0.6219 | 0.7900 | 0.7514 |
| EX2 | 0.7431 | 0.4773 | - | 0.3794 | 0.5901 | 0.5183 | 0.7592 | 0.7171 |
| EX3 | 0.7594 | 0.4667 | 0.3794 | - | 0.6353 | 0.5545 | 0.7607 | 0.7135 |
| IN1 | 0.8508 | 0.6548 | 0.5901 | 0.6353 | - | 0.6426 | 0.8471 | 0.8235 |
| IN2 | 0.7237 | 0.6219 | 0.5183 | 0.5545 | 0.6426 | - | 0.7510 | 0.7188 |
| MG | 0.8550 | 0.7900 | 0.7592 | 0.7607 | 0.8471 | 0.7510 | - | 0.8530 |
| OC | 0.8520 | 0.7514 | 0.7171 | 0.7135 | 0.8235 | 0.7188 | 0.8530 | - |

# C  Additional figures

## C.1  The UMAP visualization of the methods on PBMC10k

We have visualized the extracted features by different methods on PBMC10k dataset through UMAP, which is shown in Figure 3

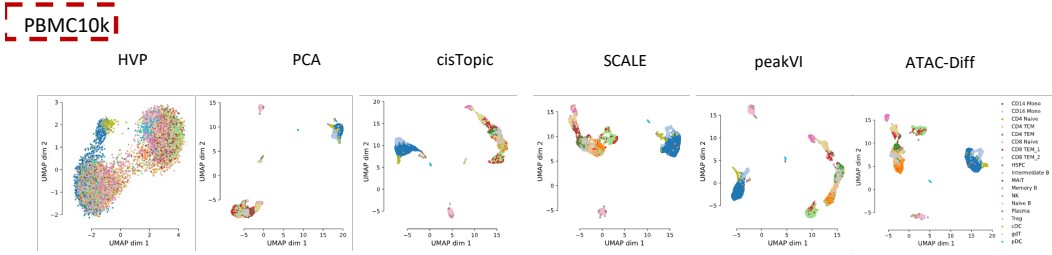

Figure 3: UMAP visualization of the highly variable peak values and extracted features from PCA, cisTopic, SCALE, PeakVI, and ATAC-Diff on PBMC10k dataset.

