# OpenReview forum: "A versatile informative diffusion model for single-cell ATAC-seq data generation and analysis"
_NeurIPS.cc/2024/Conference — NeurIPS 2024 poster_

### Official Review · Reviewer_BJyq · 2024-07-10

**Soundness:** 2
**Presentation:** 3
**Contribution:** 3
**Rating:** 4
**Confidence:** 3

**Summary:**

The paper introduces a new diffusion model ATAC-diff for scATAC-seq data generation and analysis. ATAC-diff using a latent diffusion model which is conditioned on latent auxiliary models to encode latent variables, and integrate GMM as the latent prior to capture genetic information. This paper introduces a mutual information regularization to maintain the connection between observed and latent variables. Extensive experiments show that ATAC-diff outperforms state-of-the-art models in both data generation and analysis tasks.

**Strengths:**

This paper introduces a uniform model that is applicable for multiple tasks in scATAC-seq analysis. The method section is clear and easy to follow, providing a clear explanation on auxiliary module, integration of GMM and mutual information regularization.
The paper provides an in-depth theoretical analysis, helps the audience to better understand the method.
The experiment covers a wide range of scATAC-seq analysis, including clustering, generation, denoising and imputation. The results are comprehensive and shows advantage of the new model.

**Weaknesses:**

Multiple baseline performance on the benchmark dataset does not match what was reported in previous publications. For example, [1] shows much higher PCA performance on PBMC10k dataset. This discrepancy makes me concern about the reliability of the experiment results.
If would be great to include a few benchmark dataset to help cross validate the model performance, for example Buenrostro2018.
There are top baselines that are not included in the experiment, please consider add the comparison on the latest baseline as well, such as scBasset.
[1] Yuan H, Kelley D R. scBasset: sequence-based modeling of single-cell ATAC-seq using convolutional neural networks[J]. Nature Methods, 2022, 19(9): 1088-1096.

**Questions:**

1. Explain the discrepancy on baseline performance and cross validation with previous publications.
2. Include more benchmark dataset for better cross validation of the model performance, such as Buenrostro2018.
3. Add top baselines, such as scBasset into comparison.

**Limitations:**

yes

---

> ### Author Rebuttal · Authors · 2024-08-06
>
> Thank you very much for your detailed and insightful comments on our manuscript. Your comments clearly helped a lot to improve this manuscript! We have summarized your comments and made point-by-point responses and revisions to address your concerns.
>
>
> 1. Thank you for your valuable comment. Our model outperforms the baseline models on both the analysis and generation task. The baseline models are only designed for the scATAC-seq data analysis by leveraging the approximate distribution like ZINB as the posterior. In contrast, our model incorporate both the latent distribution construction and data likelihood learning. These two strategies  can coordinate with each other, leading to the improved performance on both sides. The low dimensional cell embeddings could enhance the diffusion model to capture intrinsic high-level factors of variation present in the heterogeneous scATAC-seq data. Furthermore,  the mutual information between the cell embeddings and real data points enable ATAC-Diff to avoid ignoring the latent variables as the conditional information when utilizing ELBO as the objectivity compared to the baseline models which are based VAE framework (SCALE, PeakVI).
>
> 2. Thank you for your considerable comment. Actually, the Hematopoiesis dataset is the Buenrostro2018 dataset. We utilize Hematopoiesis to indicate the biological process insetad of Buenrostro2018 dataset. You could check the reference [1]. Jason D Buenrostro, M Ryan Corces, Caleb A Lareau, Beijing Wu, Alicia N Schep, Martin J Aryee, Ravindra Majeti, Howard Y Chang, and William J Greenleaf. Integrated single-cell analysis maps the continuous regulatory landscape of human hematopoietic differentiation. Cell, 173(6):1535–1548, 2018.
>
> 3. Thank you very much for your valuable comments. We did not compare our model with scBasset since it is only designed for single cell data analysis without any generation ability. Besides, scBasset requires the DNA sequences as the input. However, both Forebrain dataset and PBMC10K dataset do not provide the genome information. In order to further address your concern, we have reported the results of scBasset on the Hematopoiesis (Buenrostro2018) dataset.
>
> | Methods/Metrics | NMI   | ARI   | Homo  | ASW   |
> |-----------------|-------|-------|-------|-------|
> | scBasset        | 0.699 | 0.577 | 0.714 | 0.231 |
>
> While scBasset may outperform our model by leveraging supplementary genomic data for predicting chromatin accessibility, it faces limitations in capturing the distribution of single-cell data due to its reliance on restricted genomic information. Moreover, its deterministic nature restricts its utility in generating scATAC data.

---

> > ### Author Response · Authors · 2024-08-14
> >
> > Dear Reviewer,
> >
> > We would like to express our gratitude for your valuable comments again. Please do not hesitate to reach out if you have any further concerns or require additional information.
> >
> > Thanks for your attention. We are looking for your reply!

---

### Official Review · Reviewer_kixm · 2024-07-11

**Soundness:** 3
**Presentation:** 2
**Contribution:** 2
**Rating:** 7
**Confidence:** 3

**Summary:**

This submission introduces ATAC-Diff, a conditional latent diffusion model for scATAC-seq data. ATAC-Diff incorporates a few components to a basic diffusion model, including a Gaussian Mixture Model as a semantic prior over the latent variables and mutual information as a regularizer. ATAC-Diff is benchmarked on several tasks: latent representation clustering, (contitional) data generation, and denoising/imputation. ATAC-Diff is shown to do better than PeakVI and SCALE (among other methods for clustering).


[UPDATE 8/9]: Adjusting score from Reject to Accept due to clarifications.

**Strengths:**

- It seems to be the first application of a diffusion model to atac-seq data represented as cellsxpeaks
- ATAC-Diff is benchmaked against other generative models based on VAEs, including SCALE and PeakVI
- Performance of ATAC-Diff seems to be better than previous methods on 3 tasks (i.e., cell type clustering, data generation, and denoising).
- ATAC-Diff can generate unconditionally and conditionally with assistance from a VAE

**Weaknesses:**

- presentation of ATAC-seq data is not clear. The cell x peak representation is a processed version that doesn't necessarily encapsulate the full ATAC-seq data. Clarity that they are not generating ATAC-seq data but rather processed ATAC-seq data in the form of cell x binary peaks is necessary.
- Example: other approaches to analyze ATAC-seq data incorporate sequences, eg. chromBPnet and AI-TAC.
- Weak evaluations. The evaluations of cell clustering are not clear. It is not clear how ground truth was determined, given it is likely given by another computational method. The metrics are not clear what they are and their strengths and limitations. The umap visualization is not reliable. The authors state that they observe EC1, EC2, EC3 from the Forebrain dataset are close proximity in latent space. But latent space can be highly warped, rendering any Euclidean distances in umap space not meaningful.
- Generation quality task is confusing.
	- When is unconditional generation needed? Why is this a meaningful task?
	- Not clear how the metrics are calculated. What is the ground truth here?
	- Why are generated cells averaged? Shouldn't generation be assessed at the single-cell level?
	- The ATAC-Diff w.o con seems quite high. This affirms my concern that the evaluations might be weak.
- The proposed components GMM and MI regularizer are not evaluated via an ablation study. Are they even needed?
- This study treats ATAC-seq data as a cell x (binary) peaks representation, instead of fragment counts, which improves scATAC-seq analysis (see Martens et al, Nature Methods, 2023).

**Questions:**

Questions are integrated within Weaknesses (above).

**Limitations:**

This study seems interesting and their method ATAC-Diff may be an advance, but it is difficult to assess given the poor presentation. There are several weaknesses such as questionable ground truth within the benchmarks. Moreover, the task itself seems to be dated, focusing on cells x binary peaks. Moreover, the components of the proposed approach are not tested in any ablation study, bringing their purpose questionable. This approach seems to be a new method, but one of many in the growing universe of scATAC-seq data and it's not clear what impact this work will have due to these limitations.

---

> ### Author Rebuttal · Authors · 2024-08-06
>
> Thank you very much for your detailed and insightful comments on our manuscript. Your comments clearly helped a lot to improve this manuscript!
>
> 1. We clarify that in our approach, we utilize fragment counts rather than binary peaks to represent the scATAC-seq data. To alleviate any confusion, we have updated the descriptions related to the data for better clarity.
>
> "Specifically, we use fragment counts to represent the scATAC-seq data.
>
> "We compress the scATAC-seq data (fragment counts) into a lower-dimensional latent space."
>
> 2. Our model is constructed solely based on the scATAC-seq data, without incorporating DNA sequence information. Given that our model operates as a conditional generative model, we intend to integrate DNA sequence information as a guiding conditional factor in our future work. In this version, we did not use DNA sequence since some datasets do not include the genome information.
>
> 3. Ground truth cell annotations utilized in our study are extracted from the previous publication [1,2,3]. These annotations were extensively characterized through marker peaks corresponding to marker genes, supported by their distinct biological functions enriched with cell-type-specific peaks. We affirm that these cell annotations are not only reliable but also hold significant biological relevance.
>
> The metrics like NMI, ARI, and ASW, which were employed in our evaluation, are commonly utilized to assess the conservation of biological variance in latent features for single-cell dataset benchmarks. This practice is well-documented in [https://www.nature.com/articles/s41592-021-01336-8] and several other prominent single-cell methodologies [4,5,6].
>
> We acknowledge the concern that UMAP visualization can sometimes warp the space, particularly in regions with large distant gaps. Nevertheless, UMAP generally preserves distance relationships within closely-knit local subgroups that share similarities. For instance, subpopulations like EC1, EC2, and EC3, which belong to excitatory cells and are in close proximity, exhibit this preservation, aligning well with their biological characteristics.
>
> Moreover, we have computed the average latent embeddings within each cell type population and calculated the distances across different cell types. We have listed some distances due to the space limitation: EX1-EX2: 0.477, AC-EX1: 0.831. We have added other values in the Appendix.
>
> 4.1 We employ unconditional generation to assess the model's ability to capture the entirety of the data distribution. Effective modeling of the data distribution enables us to generate synthetic data for data augmentation, potentially obviating the need for sequencing additional cells and thereby saving valuable time and resources.
>
> To further address your concerns, we have refined Section 4.4.1 to provide a more detailed explanation of the unconditional task
>
> 4.2 For the cluster task, we utilize the NMI, ARI, Homo, and ASW to evaluate the performance. The calculation of metrics are formulated as follows.
>
> $NMI(A,B)=\frac{I(A,B)}{\sqrt{H(A)H(B)}}$
>
> $ARI=\frac{RI-\text{Expected}_RI}{\max{(RI)}-\text{Expected}_RI}$
>
> $Homo = 1 - \frac{H(C, Y)}{H(Y)}$
>
> $ASW = \frac{1}{N} \sum_{i=1}^{N} \frac{b_i - a_i}{\max(a_i, b_i)}$
>
> We adopt the average of all scATAC-seq data as the ground truth to calculate the SCC and PCC for the unconditional generation task. For the conditional generation, we average all single cells of the same biological cell type as the ground truth.
> For the denoising task, it is the same with conditional generation. For the imputation task, we calculate the SCC and PCC of the masked value and the imputed value.
>
> 4.3 We cannot calculate the correlation of the generated data and ground truth at the single cell level as the data is not one to one generated. The scATAC-seq technology suffers from many sources of technical noise, leading to dropout events, which means even the ground truth scATAC-seq data is also not complete.
>
> 4.4 The SCC and PCC of the conditional generation are lower than the unconditional generation since we average the SCC and PCC of different cell types for conditional generation while we calculate the SCC and PCC of the whole data for conditional generation. There are some cell types which are hard to learn due to limited data. Hence,  the PCC and SCC of such cell types are quite low, leading to the lower averaged SCC and PCC.
>
> 4.5 We have conducted the ablation study.
>
> | Clustering | NMI   | ARI   | Homo  | ASW   |NMI   | ARI   | Homo  | ASW   |NMI   | ARI   | Homo  | ASW   |
> |-----------------|-------|-------|-------|-------|-------|-------|-------|-------|-------|-------|-------|-------|
> | ATAC w.o GMM       | 0.558 | 0.438 | 0.556 | 0.202 | 0.489 | 0.297 | 0.505 | 0.222| 0.586 | 0.288 | 0.655 | 0.137 |
> | ATAC w.o MI       | 0.596 | 0.451 | 0.602 | 0.120 | 0.490 | 0.299 | 0.510 | 0.077 | 0.590 | 0.231 | 0.657 | 0.032 |
>
> | Unconditional  | SCC  | PCC   | SCC  | PCC  | SCC   | PCC  |
> |-----------------|-------|-------|-------|-------|-------|-------|
> | ATAC w.o GMM       | 0.919 | 0.991 | 0.886 | 0.949 | 0.693 | 0.710 |
> | ATAC w.o MI       | 0.916 | 0.991 | 0.904 | 0.962 | 0.704 | 0.729 |
>
> | Conditional  | SCC  | PCC   | SCC  | PCC  | SCC   | PCC  |
> |-----------------|-------|-------|-------|-------|-------|-------|
> | ATAC w.o GMM       | 0.678 | 0.768 | 0.845 | 0.910 | 0.823 | 0.911 |
> | ATAC w.o MI       | 0.681 | 0.769 | 0.848 | 0.913 | 0.831 | 0.920 |
>
> | Denoising | SCC  | PCC   | SCC  | PCC  | SCC   | PCC  |
> |-----------------|-------|-------|-------|-------|-------|-------|
> | ATAC w.o GMM       | 0.701 | 0.867 | 0.823 | 0.851 | 0.843 | 0.932 |
> | ATAC w.o MI       | 0.706 | 0.868 | 0.831 | 0.856 | 0.851 | 0.940 |
>
> | Imputing | SCC  | PCC   | SCC  | PCC  | SCC   | PCC  |
> |-----------------|-------|-------|-------|-------|-------|-------|
> | ATAC w.o GMM       | 0.710 | 0.841 | 0.887 | 0.898 | 0.833 | 0.935 |
> | ATAC w.o MI       | 0.712 | 0.845 | 0.887 | 0.901 | 0.831 | 0.931 |

---

> > ### Comment · Reviewer_kixm · 2024-08-10
> > **satisfactory response.**
> >
> > The authors have addressed my concerns.  I still don't agree with UMAP analysis as it can also warp local spaces. I will adjust my scores accordingly.

---

> > > ### Author Response · Authors · 2024-08-14
> > >
> > > Thank you very much for the comments, and we deeply appreciate your valuable suggestions. Those comments are of great value for improving the quality of the manuscript.

---

### Official Review · Reviewer_feok · 2024-07-13

**Soundness:** 3
**Presentation:** 4
**Contribution:** 3
**Rating:** 6
**Confidence:** 4

**Summary:**

Generating simulated scATAC-seq data is important for developing new methods and gaining a deeper understanding of the data. However, the simulation is challenging due to dropout and high noise in the data. Authors proposed a diffusion + VAE type of method to solve the problem. The general idea is first to use VAE to project original scATAC data to a lower embedding space. Then impose a diffusion process in the embedding space. The lower embedding space is a GMM rather than the classical isotropic Normal, which is the novelty part of the method. This configuration makes biological sense as cells can be grouped as different cell types. Due to the introduction of GMM distribution in the hidden GMM space, there exists complications in generalising the diffusion loss function. The authors have shown nice and solid derivations in the appendix. The method is then applied to three datasets on three different tasks and achieved comparable  performance as SOTA methods. It seems that authors have provided a convincing solution to the research question. While this paper is clearly written and the general idea is relatively easy to follow, it will be nice if the author could help to answer the following questions.
1. In Eq. 11, what is the parametric form of q_\phi(z|x_0)
2. In Eq. 14, what is the actual meaning of conditional information y, could you list the parameters
3. Could you provide a concrete network architecture of your network in a supplementary figure, i.e. including the tensors and their dimensions?
4. In sec 4.4.1, what is the dropout rate distribution in your simulated data, are they similar to the real data?
Depending on the answers, I may change my ratings in the future.

**Strengths:**

Generating simulated scATAC-seq data is important for developing new methods and gaining a deeper understanding of the data. However, the simulation is challenging due to dropout and high noise in the data. Authors proposed a diffusion + VAE type of method to solve the problem. The general idea is first to use VAE to project original scATAC data to a lower embedding space. Then impose a diffusion process in the embedding space. The lower embedding space is a GMM rather than the classical isotropic Normal, which is the novelty part of the method. This configuration makes biological sense as cells can be grouped as different cell types. Due to the introduction of GMM distribution in the hidden GMM space, there exists complications in generalising the diffusion loss function. The authors have shown nice and solid derivations in the appendix. The method is then applied to three datasets on three different tasks and achieved comparable  performance as SOTA methods. It seems that authors have provided a convincing solution to the research question. While this paper is clearly written and the general idea is relatively easy to follow, it will be nice if the author could help to answer the following questions.
1. In Eq. 11, what is the parametric form of q_\phi(z|x_0)
2. In Eq. 14, what is the actual meaning of conditional information y, could you list the parameters
3. Could you provide a concrete network architecture of your network in a supplementary figure, i.e. including the tensors and their dimensions?
4. In sec 4.4.1, what is the dropout rate distribution in your simulated data, are they similar to the real data?
Depending on the answers, I may change my ratings in the future.

**Weaknesses:**

Generating simulated scATAC-seq data is important for developing new methods and gaining a deeper understanding of the data. However, the simulation is challenging due to dropout and high noise in the data. Authors proposed a diffusion + VAE type of method to solve the problem. The general idea is first to use VAE to project original scATAC data to a lower embedding space. Then impose a diffusion process in the embedding space. The lower embedding space is a GMM rather than the classical isotropic Normal, which is the novelty part of the method. This configuration makes biological sense as cells can be grouped as different cell types. Due to the introduction of GMM distribution in the hidden GMM space, there exists complications in generalising the diffusion loss function. The authors have shown nice and solid derivations in the appendix. The method is then applied to three datasets on three different tasks and achieved comparable  performance as SOTA methods. It seems that authors have provided a convincing solution to the research question. While this paper is clearly written and the general idea is relatively easy to follow, it will be nice if the author could help to answer the following questions.
1. In Eq. 11, what is the parametric form of q_\phi(z|x_0)
2. In Eq. 14, what is the actual meaning of conditional information y, could you list the parameters
3. Could you provide a concrete network architecture of your network in a supplementary figure, i.e. including the tensors and their dimensions?
4. In sec 4.4.1, what is the dropout rate distribution in your simulated data, are they similar to the real data?
Depending on the answers, I may change my ratings in the future.

**Questions:**

Generating simulated scATAC-seq data is important for developing new methods and gaining a deeper understanding of the data. However, the simulation is challenging due to dropout and high noise in the data. Authors proposed a diffusion + VAE type of method to solve the problem. The general idea is first to use VAE to project original scATAC data to a lower embedding space. Then impose a diffusion process in the embedding space. The lower embedding space is a GMM rather than the classical isotropic Normal, which is the novelty part of the method. This configuration makes biological sense as cells can be grouped as different cell types. Due to the introduction of GMM distribution in the hidden GMM space, there exists complications in generalising the diffusion loss function. The authors have shown nice and solid derivations in the appendix. The method is then applied to three datasets on three different tasks and achieved comparable  performance as SOTA methods. It seems that authors have provided a convincing solution to the research question. While this paper is clearly written and the general idea is relatively easy to follow, it will be nice if the author could help to answer the following questions.
1. In Eq. 11, what is the parametric form of q_\phi(z|x_0)
2. In Eq. 14, what is the actual meaning of conditional information y, could you list the parameters
3. Could you provide a concrete network architecture of your network in a supplementary figure, i.e. including the tensors and their dimensions?
4. In sec 4.4.1, what is the dropout rate distribution in your simulated data, are they similar to the real data?
Depending on the answers, I may change my ratings in the future.

**Limitations:**

Generating simulated scATAC-seq data is important for developing new methods and gaining a deeper understanding of the data. However, the simulation is challenging due to dropout and high noise in the data. Authors proposed a diffusion + VAE type of method to solve the problem. The general idea is first to use VAE to project original scATAC data to a lower embedding space. Then impose a diffusion process in the embedding space. The lower embedding space is a GMM rather than the classical isotropic Normal, which is the novelty part of the method. This configuration makes biological sense as cells can be grouped as different cell types. Due to the introduction of GMM distribution in the hidden GMM space, there exists complications in generalising the diffusion loss function. The authors have shown nice and solid derivations in the appendix. The method is then applied to three datasets on three different tasks and achieved comparable  performance as SOTA methods. It seems that authors have provided a convincing solution to the research question. While this paper is clearly written and the general idea is relatively easy to follow, it will be nice if the author could help to answer the following questions.
1. In Eq. 11, what is the parametric form of q_\phi(z|x_0)
2. In Eq. 14, what is the actual meaning of conditional information y, could you list the parameters
3. Could you provide a concrete network architecture of your network in a supplementary figure, i.e. including the tensors and their dimensions?
4. In sec 4.4.1, what is the dropout rate distribution in your simulated data, are they similar to the real data?
Depending on the answers, I may change my ratings in the future.

---

> ### Author Rebuttal · Authors · 2024-08-06
>
> Thank you very much for your encouraging comments and the constructive advices on improving the manuscript. Your comments clearly helped a lot to improve the study. We have summarized the suggested comments (Questions) and made point-by-point responses and revisions as follows.
>
> 1. Thank you for your valuable feedback. $q_\phi(z|x_0)$ represents the amortized inference distribution, serving as an approximate variational posterior within the generative model. To achieve this parameterization, we employ an auxiliary encoder which is similar to the encoder of VAE model.
> 2. Thank you for your insightful comments. In our study, $y$ represents the conditional information, encompassing factors like cell type, tissue, and other omics data (e.g., scRNA-seq data). We have chosen to utilize cell type as the guiding conditional information for the evaluation of our model. To address your feedback, we have revised the description of $y as follows:
>
> "We conditioned the diffusion models on the latent 165 variables z and other conditional information y such as cell types, tissue, and other omics data (e.g. scRNA-seq data)."
>
> 3. Thanks for your valuable comments. We have incorporated a figure illustrating the dimensions of the tensors within the network. We will put this figure in the global response.
> 4. We sincerely appreciate your valuable insights. In our methodology, we utilize an exponential distribution where peaks exhibiting lower expression levels are more prone to dropout events compared to those with higher expression levels. This differential dropout mechanism is employed to simulate realistic dropout events in our data. We hypothesize that peaks with higher expression levels are more likely to go undetected during the sequencing process. Consequently, we ensure that a minimum of 80% of values in the simulated dataset are zero, reflecting the sparsity inherent in single-cell sequencing data. This intentional introduction of sparse noise effectively mirrors the prevalent characteristics found in actual single-cell sequencing datasets, which are often characterized by a significant number of 'missing' or 'zero' values."

---

> > ### Comment · Reviewer_feok · 2024-08-13
> >
> > I have read the rebuttal and it is ok to accept the paper.

---

> > > ### Author Response · Authors · 2024-08-14
> > >
> > > Thank you for taking the time to review our response for your comments. We would like to express our sincere gratitude for your thorough review and valuable feedback throughout the entire review process. Your insights and suggestions have been instrumental in helping us improve the quality and clarity of our work.

---

### Author Rebuttal · Authors · 2024-08-07

Thanks all the reviewers for their comments and constructive suggestions, which really help improve this manuscript. We have added the revised illustration of model in the PDF file. Moreover, we have computed the average latent embeddings within each cell type population and calculated the distances across different cell types.

| Comparison | AC    | EX1   | EX2   | EX3   | IN1   | IN2   | MG    | OC    |
|------------|-------|-------|-------|-------|-------|-------|-------|-------|
| AC         | -     | 0.8307| 0.7431| 0.7594| 0.8508| 0.7237| 0.8550| 0.8520|
| EX1        | 0.8307| -     | 0.4773| 0.4667| 0.6548| 0.6219| 0.7900| 0.7514|
| EX2        | 0.7431| 0.4773| -     | 0.3794| 0.5901| 0.5183| 0.7592| 0.7171|
| EX3        | 0.7594| 0.4667| 0.3794| -     | 0.6353| 0.5545| 0.7607| 0.7135|
| IN1        | 0.8508| 0.6548| 0.5901| 0.6353| -     | 0.6426| 0.8471| 0.8235|
| IN2        | 0.7237| 0.6219| 0.5183| 0.5545| 0.6426| -     | 0.7510| 0.7188|
| MG         | 0.8550| 0.7900| 0.7592| 0.7607| 0.8471| 0.7510| -     | 0.8530|
| OC         | 0.8520| 0.7514| 0.7171| 0.7135| 0.8235| 0.7188| 0.8530| -     |

---

### Author Response · Authors · 2024-08-12
**Inquiry about the feedback from the reviewers**

Dear ACs,

I am sorry to bother you. Could you kindly assist in verifying if the reviewers have received the rebuttal information? It has been six days since the rebuttal period concluded, and we only have two days left to discuss my response to their comments and their feedback on this manuscript.

Your prompt assistance is greatly appreciated. I am looking forward to your response. Thank you for your help.

---

> ### Comment · Area_Chair_3n8R · 2024-08-12
>
> Dear Authors,
> You are right. Most likely your rebuttal is sufficient to resolve any unclarities at this point, but if there are any further questions, they'll hopefully be posted asap so that you may still react.
> thanks for your patience,
> Area Chair

---

### Decision · Program_Chairs · 2024-09-25

**Decision:**

Accept (poster)

**Comment:**

The reviewers name the theoretical motivation and analysis of the method as well as good empirical performance as strengths. Writing is also generally speaking good, although some unclear parts were pointed out and should be clarified. Among weaknesses, some relevant comparisons were missing from the original manuscript and should be included for a more comprehensive evaluation with SOTA. Moreover, there are concerns about the evaluation metric (due to possibly misleading Euclidean distances in UMAP space).

Overall, it seems that the authors were able to address the most significant critical comments in their rebuttal (with the exception of the UMAP problem which remains an issue but, in my opinion, not a fatal one), and no further concerns were raised by the reviewers.